# RULERAG: RULE-GUIDED RETRIEVAL-AUGMENTED GENERATION WITH LANGUAGE MODELS FOR QUESTION ANSWERING

## ABSTRACT

Retrieval-augmented generation (RAG) framework has shown promising potential in knowledge-intensive question answering (QA) by retrieving external corpus and generating based on augmented context. *However, existing approaches only consider the query itself, neither specifying the retrieval preferences for the retrievers nor informing the generators of how to refer to the retrieved documents for the answers, which poses a significant challenge to the QA performance.* To address these issues, we propose Rule-Guided Retrieval-Augmented Generation with LMs, which explicitly introduces symbolic rules as demonstrations for in-context learning (RuleRAG-ICL) to guide retrievers to retrieve logically related documents in the directions of rules and uniformly guide generators to generate answers attributed by the guidance of the same set of rules. Moreover, the combination of queries and rules can be further used as supervised fine-tuning data to update retrievers and generators (RuleRAG-FT) to achieve better rule-based instruction following capability, leading to retrieve more supportive results and generate more acceptable answers. To emphasize the attribution of rules, we construct five rule-aware QA benchmarks, including three temporal and two static scenarios, and equip RuleRAG with several kinds of retrievers and generators. Experiments demonstrate that training-free RuleRAG-ICL effectively improves the retrieval quality of +89.2% in Recall@10 scores and generation accuracy of +103.1% in exact match scores over standard RAG on average across the five benchmarks, and further fine-tuned RuleRAG-FT consistently yields more significant performance enhancement. Extensive analyses indicate that RuleRAG scales well with increasing numbers of retrieved documents and exhibits generalization ability for untrained rules. Our code and benchmarks are available at https://anonymous.4open.science/r/ICLR2025_RuleRAG_ICL_FT.

## 1 INTRODUCTION

Large language models (LLMs) have achieved impressive language generation capability and excelled as knowledge learners for their well-known in-context learning (ICL) ability (Brown et al., 2020; Ouyang et al., 2022; Chowdhery et al., 2024). Despite the success, the full-parametric knowledge stored in LLMs requires substantial computational costs to keep their memory up-to-date and struggles to precisely manipulate fine-grained queries, especially in knowledge-intensive tasks (Jiang et al., 2023c; Shao et al., 2023). As complementary, RAG represents a novel framework that integrates LLMs with non-parametric information and injects the retrieved knowledge in a plug-and-play manner (Lewis et al., 2020; Dhingra et al., 2022). By explicitly decoupling the knowledge retrieval phase from the answer generation phase, RAG exhibits superior performance in many NLP tasks, such as open-domain QA (Trivedi et al., 2023) and natural language inference (Qin et al., 2023).

However, two high-level issues exist in the current RAG frameworks. First, in the retrieval phase, the imperfect retrieval component can not guarantee that the recalled information will always be the most pertinent and helpful to the queries. The reason is that the retrievers in retrieval-augmented language models (RALMs) are mostly trained on unsupervised text (Izacard et al., 2024) or trained end-to-end (Guu et al., 2020; Borgeaud et al., 2022a), leading to their insufficiency in retrieving the necessary statements for reasoning (BehnamGhader et al., 2023). Secondly, in the generation phase, the LLMs in the current RAG are not specifically informed of how to exploit noisy retrieved

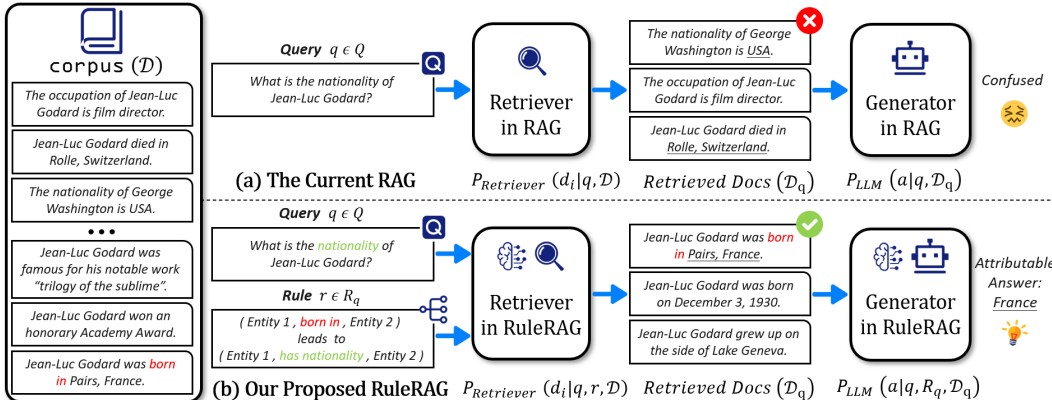

Figure 1: (a) Without the help of rules, the current RAG can only retrieve relevant documents at the shallow semantic level, rather than the overall semantics of the query, and thus get confused in answering. (b) Guided by the rule $r$ related to the query, our proposed RuleRAG first retrieves supportive documents that are logically related to the query and then attributes the correct answer, "France".

content properly, since relationships between a wide range of facts are rarely explicitly "pointed out" and "supervised" in the pre-training corpora of LLMs. For example, REPLUG (Shi et al., 2023) and In-Context RALM (Ram et al., 2023) fuse off-the-shelf LLMs with generic retrievers and treat LLMs as black boxes. Even if answered correctly, they still lead to implicit attribution processes that are difficult to explain and verify. Therefore, these RAG frameworks are neither inherently trained to retrieve along reasonable retrieval directions nor organically attribute retrieved content to answers.

In contrast to these existing RAG frameworks, we observe that widespread logical rules can explicitly guide people to accomplish a given task. For example, for the addition of large numbers, a math problem, humans can easily solve the addition of any two numbers after learning the rules of column addition. To implement rule-based calculating, Hu et al. (2024) instructs transformers to cite basic addition rules by explicitly fine-tuning transformers with them. While answering factual queries by means of retrieve-then-read in our task, human experts spontaneously search for relevant information as intermediate results following a priori rules before answering and refer to these rules again while deciding final answers (Shnarch et al., 2020). As shown in Figure 1 (a), if we ask What is the nationality of Jean-Luc Godard? and no directly relevant information is contained in the corpus $\mathcal{D}$, the current retrievers depend on shallow representations and lead to recall many word-level similar documents which involve Jean-Luc Godard or Nationality. In this case, the retrieved information does not contribute anything semantically to answering. In fact, we know a prior the rule that if someone is born in a certain country, there is a high probability that his (her) nationality is also that country. Therefore, we can leverage this rule to conduct more effective and accurate retrieval (Huang et al., 2023) and offer documents that can better support question answering (Figure 1 (b)). Similarly, in the stage of generating answers, since we input multiple documents, LLMs may get confused by a large amount of irrelevant information. If LLMs are explicitly instructed to incorporate the retrieved content and the queries through rules, they will better attribute answers as humans expect.

Upon the above motivation, we propose Rule-Guided Retrieval-Augmented Generation (RuleRAG), a new QA approach, that enables to recall documents logically supporting queries explicitly in the directions of rules and generates the final answers based on retrieved information and attributable rules. Compared to standard RAG, training-free RuleRAG-ICL requires the introduction of rules with high confidence in the input sides of the retrievers and generators, aiming to guide the document retrieval and answer attribution processes. Moreover, to cultivate and boost the rule-following ability of LMs, we further propose RuleRAG-FT, which retrofits retrievers and generators via our designed rule-guided fine-tuning (RGFT). Specifically, for rule-guided retriever fine-tuning (RGFT-retriever), we construct pairs of queries and rules as input with the training labels of oracle retrieval examples; for rule-guided generator fine-tuning (RGFT-generator), we construct pairs of retrieved results, rules and queries as input with the training labels of golden answers. In practice, we find that our obtained documents are highly compatible with queries and the generated answers are fairly targeted with ground truths since the introduced rules improve the ability of LMs to organize causal relationships between facts (Peng et al., 2024). In summary, both RuleRAG-ICL and RuleRAG-FT can associate the retrieval stage with the generation stage, achieving better recall of information and better answer accuracy.

To demonstrate the effectiveness of RuleRAG, we newly construct five rule-aware QA benchmarks, where the queries require answering temporal or static real-world factual questions and the answers are the concrete entity names (Figure 1 is an example). It has been prone that the answers are hard to directly predict based merely on queries due to limited storage capacity (Petroni et al., 2019; Dhingra et al., 2022) and factual hallucinations (Zhang et al., 2023) of LLMs. Therefore, each benchmark offers a fact corpus which serves as the external data pool and contain documents possibly related to queries. Our experiments show that, under several retrieval and generation configurations, RuleRAG-ICL offers considerable performance gains with the individual guidance of rules by in-context learning and RuleRAG-FT achieves further improvements by combining the fine-tuned retrievers and generators. Extensive comparative studies and analyses confirm the superiority of symbolic rules and show the effectiveness of RuleRAG across various types of retrievers and generators. Moreover, RuleRAG-FT can be extrapolated to unseen rules without retraining owing to the transferable rule utilizing abilities during retrieving and generating enhanced by our designed rule-guided fine-tuning (RGFT).

## 2 Newly Constructed Rule-aware QA Benchmarks

**Rule bank $\mathcal{R}$.** A huge amount of world knowledge, including static facts and temporal events, has been stored in static KGs and temporal KGs (Jiang et al., 2023b). In the static scenario, several different relations can be simultaneously established between two entities. In the temporal scenario, two entities can interact multiple times at different timestamps. Hence, if relation $r_1$ (rule body) can logically explain the occurrence of relation $r_2$ (rule head) between entities, we represent this relevance as rule $r$ in a natural language form: *[Entity 1, $r_1$, Entity 2] leads to [Entity 1, $r_2$, Entity 2]*. We leverage the classical rule mining algorithm AMIE3 (Lajus et al., 2020) for static KGs and TLogic (Liu et al., 2022) for temporal KGs. The frequently co-occur relations form rules with high confidence (Liao et al., 2024) and we transform them to the above text string form. All these individual rules comprise our rule bank $\mathcal{R}$, which will be consistently leveraged in the training and inferring process of RuleRAG.

**Test dataset $\mathbb{Q}$.** To avoid skewed entity distribution, we include links with both popular and long-tail entities in KG test sets and adjust their numbers to achieve balance. The remaining links are converted into queries with tail entities in these links as ground truths. Different from PopQA (Mallen et al., 2023) with more low-popularity entities from Wikidata, our benchmarks consider entities in uniform distribution from five knowledge bases, aiming to show the more general effectiveness of our method.

**Corpus $\mathcal{D}$ and fine-tuning datasets, $\mathcal{F}_R$ and $\mathcal{F}_G$.** Different from EntityQuestions (Sciavolino et al., 2021), we linearize the links in KG training sets into documents by concatenating entity, relation and time, forming concise and distinct factoids in $\mathcal{D}$, which serves as the retrieval source of RuleRAG. For RGFT, we split valid sets of KGs into two disjoint parts and convert the KG links of both parts into queries: one part is for queries in the fine-tuning datasets $\mathcal{F}_R$ for retrievers and the other part is for queries in the fine-tuning datasets $\mathcal{F}_G$ for generators. Specifically, we search the corresponding oracle document examples from $\mathcal{D}$ for each query-rule pair by entity name and relation-matching heuristics and take them as the golden training labels of the retrievers. Subsequently, we leverage the fine-tuned retrievers to retrieve relevant documents for each query in $\mathcal{F}_G$ and create fine-tuning instructions for generators by combining retrieval results, rules and queries, with golden answers as supervision.

The statistics of our newly constructed QA benchmarks are in Table 1. Benchmarks with temporal queries, named RuleQA-I, RuleQA-Y and RuleQA-W, are constructed based on three temporal KGs, ICEWS14 (García-Durán et al., 2018), YAGO (Mahdisoltani et al., 2013) and WIKI (Leblay & Chekol, 2018). Benchmarks with static queries, named RuleQA-F and RuleQA-N, are constructed based on two static KGs, FB15K-237 (Toutanova & Chen, 2015) and NELL-995 (Xiong et al., 2017).

Table 1: The statistics of the constructed five rule-aware QA benchmarks in this paper. $|\mathcal{R}|$, $|\mathcal{D}|$, $|\mathcal{F}_R|$, $|\mathcal{F}_G|$ and $|\mathbb{Q}|$ represent the numbers of rules in rule banks, documents in corpus, retriever fine-tuning query-documents pairs, generator fine-tuning query-answer pairs and test queries, respectively.

| Benchmarks | $|\mathcal{R}|$ | $|\mathcal{D}|$ | $|\mathcal{F}_R|$ | $|\mathcal{F}_G|$ | $|\mathbb{Q}|$ | Query Type |
|---|---|---|---|---|---|---|
| RuleQA-I | 557 | 77,508 | 6,594 | 7,440 | 1,559 | Temporal |
| RuleQA-Y | 99 | 243,633 | 28,153 | 22,765 | 1,864 | Temporal |
| RuleQA-W | 78 | 584,364 | 50,996 | 62,375 | 2,065 | Temporal |
| RuleQA-F | 367 | 49,088 | 8,082 | 9,645 | 1,233 | Static |
| RuleQA-N | 234 | 18,177 | 4,351 | 4,764 | 815 | Static |

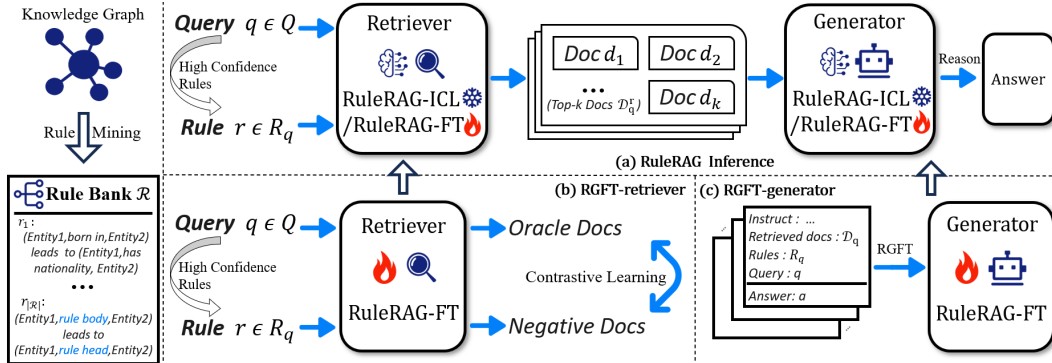

Figure 2: The framework of our proposed RuleRAG, including RuleRAG-ICL and RuleRAG-FT. RuleRAG-ICL relies on in-context learning with the guidance of rules. RuleRAG-FT involves fine-tuning retrievers and generators ahead. (a) The unified inference process of RuleRAG. (b) Rule-guided retriever fine-tuning (RGFT-retriever). (c) Rule-guided generator fine-tuning (RGFT-generator).

## 3 PROPOSED METHOD: RULERAG

In this section, we present details of our proposed novel rule-guided retrieval-augmented generation with LMs (RuleRAG) for solving the task of knowledge-intensive factual queries. Notably, RuleRAG includes training-free RuleRAG-ICL and fine-tuned RuleRAG-FT. First, we prompt RuleRAG-ICL with queries and rules for in-context learning during retrieving and inferring. The rules are aimed to guide retrievers to recall logically supportive documents and guide generators to predict attributable answers. Then, RuleRAG-FT further fine-tunes the retrievers and generators to explicitly enhance their rule-following ability by our introduced rule-guided fine-tuning (RGFT), where we leverage the queries combining rules as fine-tuning data and the ground truth answers as supervision data. The inferring process of RuleRAG-FT is the same as RuleRAG-ICL.

### 3.1 RULERAG-ICL

Figure 2 (a) illustrates the inference flow of RuleRAG-ICL. Given a query $q \in Q$, we select a few related rules $R_q$ from $\mathcal{R}$. Specifically, we first recognize the relation in the query $q$ and then retrieve the rules with this relation as the rule head, forming the guidance rules $R_q$ for $q$. We append $q$ with one rule $r \in R_q$ once at a time to avoid conflict and conduct rule-guided retrieval in the corpus $\mathcal{D}$ to obtain the top-k relevant documents $\mathcal{D}_q^r$, where $q$ provide the retrieval content and $r$ provide retrieval directions. Finally, $\mathcal{D}_q^r$ from all rules in $R_q$ are assembled to produce the final retrieval results $\mathcal{D}_q$, and RuleRAG-ICL conditions on the query $q$, rules $R_q$ and documents $\mathcal{D}_q$ to reason the answer $a$.

**Rule-guided retriever (RG-retriever).** Since each rule $r \in R_q$ stands for a unique retrieval logic, we retrieve all query-rule pairs $(q,r)$ individually to avoid the conflict of different rules. Specifically, the retriever calculates a relevant score $s(d_i, q \circ r)$ between a $(q,r)$ pair and every document $d_i \in \mathcal{D}$: $s(d_i, q \circ r) = \mathbf{E}_d(d_i) \cdot \mathbf{E}_q(q \circ r)$, where $\circ$ denotes sequence concatenation, $\cdot$ is dot product, $\mathbf{E}_d$ is the document encoder and $\mathbf{E}_q$ is the query encoder. To stay within the context window size limit of LLMs, we select the top-k scored documents, denoted as $\mathcal{D}_q^r$ ($r \in R_q$), for each $(q,r)$ pair and combine all $\mathcal{D}_q^r$ as the final retrieval results $\mathcal{D}_q$ for query $q$. This process is formalized as follows:

$$\mathcal{D}_q = \bigcup_{r \in R_q} \mathcal{D}_q^r, \ (R_q \subsetneq \mathcal{R}); \quad \mathcal{D}_q^r = \arg \underset{d_i \in \mathcal{D}}{\text{top-k}} \ s(d_i, q \circ r), \ (r \in R_q). \quad (1)$$

**Rule-guided generator (RG-generator).** After recalling $\mathcal{D}_q$, we construct an instruction to prompt LLMs to generate the final answer $a$. Different from the widely used case-based prompts (Wei et al., 2024), we do not let LLMs learn the reasoning mode implicitly from examples, but directly inform LLMs of $R_q$ as the attribution mechanisms and make LLMs answer the query $q$ explicitly according to the $\mathcal{D}_q$. Under the guidance of rules, the probability of outputting $a$ can be approximated as follows:

$$P(a \mid q) = P(a \mid q, \mathcal{R}, \mathcal{D}) \approx P_{LLM}(a \mid \text{INSTRUCTION}(q, R_q, \mathcal{D}_q)), \quad (2)$$

where $P_{LLM}()$ is the generation probability of LLMs and INSTRUCTION() is the instruction prompt. The simplified form of the instruction is in Figure 2 (c) and the detail is given in the Appendix A.9.

## 3.2 RULERAG-FT

The overview of our proposed rule-guided retriever and generator fine-tuning in RuleRAG-FT are illustrated in Figure 2 (b) and (c), respectively. For *rule-guided retriever fine-tuning* (RGFT-retriever), we update the LM encoders in a contrastive learning objective (Chen et al., 2020) and train over supervised fine-tuning data $\mathcal{F}_R$ provided in our constructed benchmarks, where inputs are the queries plus rules and supervised labels are heuristic oracle documents. Compared with retrievers employed with simple retrieval principles, our fine-tuned retrievers can recall more relevant results, aligned with the preferences of the rules. For *rule-guided generator fine-tuning* (RGFT-generator), we adopt the supervised instruction-tuning objective (Iyer et al., 2023; Chung et al., 2024) while combining each query $q$ with two components: retrieved documents $\mathcal{D}_q$ from the retrieval phase and the same set of rules $\mathrm{R}_q$ consistent with the retrieval phase. The rules introduced in the RGFT-generator train LLMs on how to optimally attribute from the retrieved context into answers by following rules, making RuleRAG leverage the fine-tuned retrievers more rationally. Experiments show our proposed RGFT can further guarantee and boost the retrieval quality and answering accuracy of RuleRAG-FT than RuleRAG-ICL.

**Rule-guided retriever fine-tuning (RGFT-retriever).** We utilize two main types of retrievers: sparse retrievers and dense retrievers. As the sparse retriever, we use Pyserini [1] to implement the standard training-free BM25 (Robertson & Zaragoza, 2009), which relies on word-level frequencies. As the dense retrievers, we adopt the dual-encoder based retriever architecture, such as DPR [2] and SimCSE [3]. We freeze the document encoder and tune the query encoder for high retrieval efficiency (Lewis et al., 2020). Given a $((q, r), \mathcal{D}_o)$ pair in the fine-tuning data $\mathcal{F}_R$ where $\mathcal{D}_o$ serve as the oracle documents, each $d_i^+ \in \mathcal{D}_o$ is a positive learning example while each in-batch $d_j^- \notin \mathcal{D}_o$ is a negative example. We train the retrievers in an in-batch contrastive training fashion with the following loss function $\mathcal{L}_q^r$:

$$\mathcal{L}_q^r = -\log \frac{\exp(s(d_i^+, q \circ r))}{\exp(s(d_i^+, q \circ r)) + \sum_{d_j^- \epsilon \mathcal{B}/\mathcal{D}_o} \exp(s(d_j^-, q \circ r))}, \qquad (3)$$

where $\mathcal{B}$ represents the documents for all the queries in one training batch. $\mathcal{D}_o$ represents oracle documents for the query and $\mathcal{B}/\mathcal{D}_o$ represents its in-batch negative examples. Retrievers are fine-tuned over $\mathcal{F}_R$. The training goal of RGFT-retriever is to minimize the overall loss $\mathcal{L} = \sum_{((q,r),\mathcal{D}_o) \in \mathcal{F}_R} \mathcal{L}_q^r$.

**Rule-guided generator fine-tuning (RGFT-generator).** From RuleRAG-ICL, we find LLMs have a certain in-context learning ability to understand the rules. For greater model efficiency and control of the output, we fine-tune our generators in RuleRAG-FT and further enhance the proficiency of LLMs to attribute accurate answers following the instruction prompt. Formally, the designed instruction contains three parts: the relevant facts $\mathcal{D}_q$ retrieved by retrievers fine-tuned above, the rules $\mathrm{R}_q$ guiding attributable retrieval logics and the original query $q$. The instruction prompt remains the same during the fine-tuning of generators and inferring of RuleRAG to keep a similar knowledge distribution.

In practice, for open-source LLMs, we utilize the few-shot instruction fine-tuning strategy considering the following two aspects. First, our introduced rules reform the data-centric training to the alignment of task-centric abilities, i.e., it can be viewed as a reasoning task based on the guidance of rules (Zhou et al., 2023) and our training aim is to learn to use them. Secondly, tuning all the data in $\mathcal{F}_G$ is prohibitive in time. We randomly select a fixed number of samples from $\mathcal{F}_G$ to conduct few-shot tuning (2048 samples in our practice). Our experiments show the effectiveness and generalization of RuleRAG-FT although the samples can not cover all the rules. For closed-source LLMs, we perform 3-shot prompts as an empirical substitute of fine-tuning (Dai et al., 2023) due to the unavailable parameters. Specifically, we randomly select three $((q, \mathcal{D}_q, \mathrm{R}_q), a)$ pairs from $\mathcal{F}_G$ as fixed examples in the prompts, making up the in-context augmentation. The detailed prompts are in the Appendix A.9.

## 4 EXPERIMENTAL SETTINGS

### 4.1 SETUP OF RULERAG

For our proposed RuleRAG-ICL, in addition to adding rule guidance to both retrievers and generators (RG-retriever + RG-generator), we also add rule guidance only to the retrieval stage (RG-retriever +

---

[1] https://github.com/castorini/pyserini

[2] https://github.com/facebookresearch/DPR

[3] https://github.com/princeton-nlp/SimCSE

generator), trying to prove that introducing rules in two stages can both contribute to the performance. For our proposed RuleRAG-FT, the complete method involves retrievers and generators with RGFT. The ablation study shows both of them are individually beneficial to the results. To emphasize the contribution of rules, we introduce several variants of RuleRAG-FT. The SSFT in Table 2 represents the standard supervised fine-tuning following the vanilla manner, where the fine-tuning instruction consists only of the queries and retrieved documents without rules. Note that whether or not the inputs are added with rules during inference is consistent with how the models are fine-tuned during training.

## 4.2 BASELINES

Given that LLMs have pre-trained with lots of world knowledge, we report the performance of directly using LLMs as answer predictors without retrieval (Standard Prompting in Table 2) for basic performance reference (Ouyang et al., 2024). Additionally, we compare RuleRAG with a wide range of baselines based on retrieval-augmented generation (RAG). We instantiate the widespread RAG framework using off-the-shelf LLMs and retrievers with queries as input, standing for the standard RAG methods (Standard RAG in Table 2 and 3). Chain-of-thought (CoT) methods, verify-and-edit (VE; Zhao et al. (2023)) and chain-of-knowledge (CoK; Li et al. (2024)) correct LLM outputs independently and sequentially respectively by leveraging external knowledge sources. Following their implementation, we initialize the knowledge sources as our corpus $\mathcal{D}$ for a fair comparison and use 3-shot CoT prompts.

## 4.3 EVALUATION METRICS

For the retrieval stage, the quality of retrieved documents is critical for downstream queries and is usually measured by Recall@k (Karpukhin et al., 2020), indicating whether the top-k blocks contain targeted information. For our task, we calculate Recall@k (**R@k**,%) by checking whether the correct answer to the given query is contained in the retrieved top-k documents. The higher R@k, the more potentially useful retrievers are for generators. For the generation stage, the quality of answers is measured by Exact Match (**EM**,%) and Token F1 (**T-F1**,%), which are widely recognized in QA performance evaluation (Zhu et al., 2021). For EM, an answer is deemed correct if its normalized form corresponds to any acceptable answer in the provided ground truth lists. T-F1 treats the answers and ground truths as bags of tokens and computes the average token-level overlap between them (Li et al., 2023b).

## 5 EXPERIMENTAL RESULTS

### 5.1 MAIN RESULTS

Table 2 shows the overall experimental results in the five rule-aware QA benchmarks detailedly and provides a comprehensive comparison between our proposed RuleRAG-ICL, RuleRAG-FT and all the baselines, under the concrete instantiation of DPR (Karpukhin et al., 2020) and LLAMA2_7B (Touvron et al., 2023) as retrievers and generators. As a baseline without retrieval, LLAMA2_7B using standard prompting can only refer to the knowledge it acquired during pre-training. Unsurprisingly, we notice that Standard Prompting (LLAMA2_7B) yields the worst relative and absolute results in all the five benchmarks, revealing that parametric knowledge in LLMs makes it hard to answer our factual queries. Furthermore, the results of Standard Prompting avoid the concern that the performance improvement of subsequent experiments comes from intrinsic knowledge in LLMs. This also gives a side note to the challenges of our constructed five benchmarks and motivates the introduction of rules.

The CoT-based methods, VE and CoK, use the rationales corrected by the retrieved knowledge to enhance the factual correctness of LLMs. From their results, it is evident that although they happen to succeed in modifying some answers by using rationales, they still fail to capture the logical relationships between the broader set of facts. The Standard RAG framework has better performance than the above non-retrieval or self-verifying methods, highlighting the importance of retrieved documents for knowledge-intensive queries. However, their low performance is still unsatisfactory, suggesting that their principles of retrieval and generation are weak and leave much to be desired. In the experiments, we illustrate that the performance can be further improved under the guidance of rules from two perspectives: through in-context learning (ICL) in RuleRAG-ICL and through RGFT in RuleRAG-FT.

For RuleRAG-ICL (RG-DPR + LLAMA2_7B), introducing rules in the retrieval stage alone enhances the recall performance of the retriever and further improves the answer accuracy of the original

Table 2: Performance comparison of RuleRAG-ICL and RuleRAG-FT with their variants and baselines. RG-DPR and RG-LLAMA2_7B represent rule-guided DPR and rule-guided LLAMA2_7B in RuleRAG-ICL. RGFT represents rule-guided fine-tuning in RuleRAG-FT. SSFT represents standard supervised fine-tuning (Section 4.1). Standard Prompting does not have a retrieval stage, so there is no R@10. VE and CoK involve multiple search objects, which change several times, so the R@10 loses reference value. **The best performance of RuleRAG-ICL and RuleRAG-FT are in bold.**

| | Architecture | | RuleQA-I | | | RuleQA-Y | | | RuleQA-W | | | RuleQA-F | | | RuleQA-N | | |
|---|---|---|---|---|---|---|---|---|---|---|---|---|---|---|---|---|---|
| | Retriever | Generator | R@10 | EM | T-F1 | R@10 | EM | T-F1 | R@10 | EM | T-F1 | R@10 | EM | T-F1 | R@10 | EM | T-F1 |
| Standard Prompting | None | LLAMA2_7B | - | 1.5 | 19.4 | - | 0.4 | 12.4 | - | 1.5 | 27.7 | - | 1.0 | 24.9 | - | 0.1 | 10.4 |
| VE (3-shot) | DPR | LLAMA2_7B | - | 3.1 | 10.7 | - | 0.8 | 6.5 | - | 4.2 | 25.2 | - | 7.4 | 12.7 | - | 4.8 | 14.1 |
| CoK (3-shot) | DPR | LLAMA2_7B | - | 4.0 | 12.5 | - | 1.9 | 10.4 | - | 5.7 | 29.0 | - | 9.8 | 18.7 | - | 7.4 | 21.6 |
| Standard RAG | DPR | LLAMA2_7B | 14.1 | 5.2 | 24.4 | 3.8 | 2.6 | 18.5 | 7.4 | 4.8 | 35.8 | 18.9 | 11.0 | 33.1 | 19.3 | 9.8 | 29.6 |
| RuleRAG-ICL | RG-DPR | LLAMA2_7B | 24.2 | 5.5 | 25.1 | 6.6 | 4.3 | 19.2 | 22.6 | 10.9 | 37.1 | 29.9 | 13.1 | 33.1 | 26.5 | 11.1 | 30.6 |
| | RG-DPR | RG-LLAMA2_7B | **24.2** | **9.8** | **29.1** | **6.6** | **6.1** | **20.9** | **22.6** | **12.7** | **39.1** | **29.9** | **19.0** | **35.7** | **26.5** | **15.2** | **32.8** |
| RuleRAG-FT | RGFT-DPR | RGFT-LLAMA2_7B | **45.1** | **20.5** | **38.9** | **55.7** | **44.6** | **41.6** | **49.9** | **41.6** | **47.5** | **95.1** | **34.9** | **48.4** | **92.5** | **42.0** | **57.9** |
| *Rule Ablation* | | | | | | | | | | | | | | | | | |
| | SSFT-DPR | RGFT-LLAMA2_7B | 38.4 | 18.7 | 38.4 | 46.5 | 41.5 | 38.4 | 39.3 | 36.9 | 42.4 | 79.0 | 31.5 | 47.3 | 80.7 | 42.0 | 55.2 |
| variants of RuleRAG-FT | RGFT-DPR | SSFT-LLAMA2_7B | 45.1 | 15.3 | 27.5 | 55.7 | 43.7 | 33.2 | 49.9 | 29.4 | 34.1 | 95.1 | 14.2 | 29.6 | 92.5 | 29.8 | 42.4 |
| | SSFT-DPR | SSFT-LLAMA2_7B | 38.4 | 13.8 | 27.3 | 46.5 | 37.4 | 33.8 | 39.3 | 28.8 | 34.3 | 79.0 | 12.0 | 27.1 | 80.7 | 27.5 | 41.9 |
| *RGFT Ablation* | | | | | | | | | | | | | | | | | |
| variants of RuleRAG-FT | RG-DPR | RGFT-LLAMA2_7B | 24.2 | 13.3 | 37.7 | 6.6 | 13.9 | 25.6 | 22.6 | 14.7 | 30.5 | 29.9 | 21.6 | 36.7 | 26.5 | 15.4 | 34.9 |
| | RGFT-DPR | RG-LLAMA2_7B | 45.1 | 14.2 | 33.1 | 55.7 | 33.9 | 36.5 | 49.9 | 38.7 | 43.4 | 95.1 | 33.5 | 41.9 | 92.5 | 37.2 | 47.6 |

LLAMA2_7B. RuleRAG-ICL (RG-DPR + RG-LLAMA2_7B) consistently surpasses Standard RAG across various metrics (+9.3 in R@10, +5.9 in EM and +3.2 in T-F1 on average absolute performance over all five benchmarks), achieving the improved performance. This confirms the sub-optimal ability of the current RAG and the effectiveness of our proposed dual rule-guided retriever and generator. For RuleRAG-FT, our proposed RGFT can amazingly improve performance by a significant margin (+45.7 in R@10, +24.2 in EM and +15.3 in T-F1 compared to the best performance of RuleRAG-ICL). *To further corroborate that these gains are due to the introduced rules*, we first isolate the key component, rules, from fine-tuning data $\mathcal{F}_R$ for RGFT, to form the standard supervised fine-tuning (SSFT) (*Rule Ablation* in Table 2) and then isolate the impact of the fine-tuned generator from the fine-tuned retriever in RuleRAG-FT (*RGFT Ablation* in Table 2). *RGFT Ablation* shows both RGFT-DPR and RGFT-LLAMA2_7B are beneficial when used individually, implicitly suggesting that the two phases do not depend on each other. Moreover, *Rule Ablation* shows when we no longer leverage rules to explicitly inform the retrievers of the retrieval directions (SSFT-DPR) or how LLMs should correctly utilise the retrieved documents while fine-tuning (SSFT-LLAMA2_7B), our recall and generation performances show varying degrees of degradation compared to RuleRAG-FT. This further clarifies the great assistance of rules on our method's ability to answer knowledge-intensive queries.

## 5.2 RESULTS OF MORE LLMS

To test the generalizability to more generators in RuleRAG-ICL and RuleRAG-FT, we evaluate how different LLMs affect the performance in Table 3. We experiment with three more open-source LLMs: ChatGLM2_6B (Du et al., 2022), Mistral_7B_v0.2 (Jiang et al., 2023a), LLAMA2_13B (Touvron et al., 2023), and a closed-source LLM, GPT-3.5-Turbo [4], which can be called through OpenAI API.

First, consistent with the conclusions for the LLAMA2_7B in Table 2, the results in Table 3 show RuleRAG is effective under various kinds of LLMs. RuleRAG-ICL and RuleRAG-FT improve the overall performance of Standard RAG across all benchmarks and LLMs, demonstrating the validity and universality of rules. RuleRAG-FT consistently outperforms RuleRAG-ICL. Secondly, for LLAMA2 as generators, Standard RAG, RuleRAG-ICL and RuleRAG-FT with the 13B model always outperform their 7B counterparts, indicating that the introduced rules can provide better guidance when using larger models with the same LLM architecture. Thirdly, take LLAMA2_13B as an instance, the EM results of RuleRAG-ICL with GPT-3.5-Turbo are better than RuleRAG-ICL with LLAMA2_13B for the more massive model parameters, however, the EM results of RuleRAG-FT with LLAMA2_13B are better than RuleRAG-FT with GPT-3.5-Turbo in three of the five benchmarks. This phenomenon illustrates that RGFT is fairly effective and necessary for lightweight LLMs, making RuleRAG-FT much cheaper than off-the-shelf big LLMs for LLM deployment and application.

---

[4]https://openai.com/index/gpt-3-5-Turbo-fine-tuning-and-api-updates/

Table 3: The performance of RuleRAG-ICL and RuleRAG-FT with different LLMs as generators. The retriever is fixed as DPR. We omit R@10 since it has been given in detail in Table 2. We use 3-shot prompts for the closed-source GPT-3.5-Turbo to replace RGFT due to its unpublished parameters.

| Architecture | | | RuleQA-I | | RuleQA-Y | | RuleQA-W | | RuleQA-F | | RuleQA-N | |
|---|---|---|---|---|---|---|---|---|---|---|---|---|
| | Retriever | Generator | EM | T-F1 | EM | T-F1 | EM | T-F1 | EM | T-F1 | EM | T-F1 |
| Standard RAG | DPR | ChatGLM2_6B | 0.0 | 5.1 | 0.3 | 7.8 | 0.3 | 18.1 | 0.1 | 21.0 | 0.0 | 0.0 |
| RuleRAG-ICL | RG-DPR | RG-ChatGLM2_6B | 2.5 | 16.9 | 1.3 | 13.7 | 3.0 | 26.7 | 10.8 | 27.3 | 0.5 | 1.7 |
| RuleRAG-FT | RGFT-DPR | RGFT-ChatGLM2_6B | 7.3 | 21.2 | 42.2 | 35.2 | 23.5 | 30.5 | 19.2 | 29.8 | 25.6 | 25.6 |
| Standard RAG | DPR | Mistral_7B_v0.2 | 1.6 | 13.8 | 0.7 | 11.9 | 1.3 | 21.8 | 3.1 | 22.4 | 0.9 | 1.5 |
| RuleRAG-ICL | RG-DPR | RG-Mistral_7B_v0.2 | 3.1 | 20.0 | 4.5 | 23.4 | 34.2 | 40.7 | 6.4 | 28.6 | 4.2 | 16.6 |
| RuleRAG-FT | RGFT-DPR | RGFT-Mistral_7B_v0.2 | 22.6 | 34.9 | 49.2 | 47.3 | 35.5 | 45.2 | 53.7 | 48.9 | 50.9 | 62.6 |
| Standard RAG | DPR | LLAMA2_13B | 6.1 | 25.9 | 4.0 | 20.2 | 6.0 | 28.6 | 12.6 | 34.9 | 10.2 | 31.6 |
| RuleRAG-ICL | RG-DPR | RG-LLAMA2_13B | 10.0 | 30.0 | 6.5 | 23.7 | 14.1 | 43.4 | 20.5 | 36.9 | 18.2 | 36.1 |
| RuleRAG-FT | RGFT-DPR | RGFT-LLAMA2_13B | 22.0 | 39.8 | 46.6 | 47.9 | 42.3 | 48.1 | 45.6 | 49.6 | 42.1 | 55.6 |
| Standard RAG | DPR | GPT-3.5-Turbo | 9.0 | 29.1 | 4.8 | 25.9 | 6.9 | 31.5 | 25.7 | 24.5 | 16.0 | 43.3 |
| RuleRAG-ICL | RG-DPR | RG-GPT-3.5-Turbo | 12.2 | 30.3 | 9.9 | 28.1 | 16.4 | 33.7 | 37.9 | 32.1 | 27.5 | 50.6 |
| RuleRAG-FT | RGFT-DPR | RG-GPT-3.5-Turbo (3-shot) | 15.7 | 33.8 | 40.1 | 32.8 | 38.9 | 35.4 | 72.4 | 34.1 | 68.1 | 56.1 |

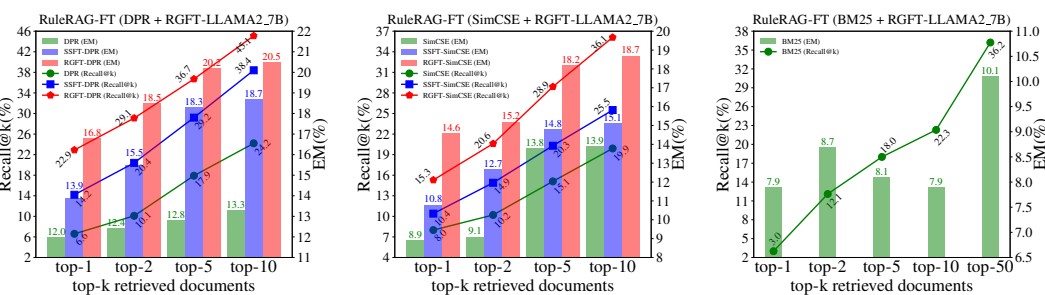

Figure 3: The Reacll@k and EM performance of RuleRAG-FT in RuleQA-I with different numbers of retrieved documents and under multiple circumstances: three settings in DPR (DPR, SSFT-DPR and RGFT-DPR), three settings in SimCSE (SimCSE, SSFT-SimCSE and RGFT-SimCSE) and one setting in BM25. The generator is kept as RGFT-LLAMA2_7B. Horizontal numbers over the pillars represent EM for bar charts and slanted numbers around the lines represent Recall@k for line charts.

## 5.3 RESULTS OF MORE RETRIEVERS

### 5.3.1 MORE RETRIEVERS FOR RULERAG-FT

In Figure 3, we initialize RuleRAG-FT with more retrievers: dense retrievers DPR (Karpukhin et al., 2020), SimCSE (Gao et al., 2021) and training-free sparse retriever BM25 (Robertson & Zaragoza, 2009), and we use several retrieval configurations: retrievers without fine-tuning or with SSFT/RGFT while recalling different numbers of top-scored documents. Before fine-tuning, the Recall@k and EM performance of the three retrievers are comparable and each has its own performance, with no obvious advantages or disadvantages. For instance, comparing the three retrievers side-by-side, DPR has the best Recall@10 and SimCSE has the best EM under top-10 documents before fine-tuning.

After fine-tuning, DPR consistently outperforms SimCSE and RGFT consistently outperforms SSFT. Specifically, under considering top-scored documents with the same k, for the two trainable dense retrievers, the RGFT version recalls more relevant information (Recall@k) than the SSFT version by a large margin, demonstrating the generality of the proposed RGFT across different retrievers. As a result, the EM scores of the generated answers are better when higher-quality documents from retrievers are provided. Moreover, when the retrievers and generators are applied with RGFT, RuleRAG-FT shows substantial performance gains, even with the retrieval number limited to top-1. For DPR and SimCSE, as we include more documents, the Recall@k and EM scores increasingly improve. This shows that leveraging rules to guide the retrieval and generation processes builds a bridge between queries and answers since rules provide retrieval directions and attributable mechanisms. For BM25, although Recall@k keeps increasing, EM experiences a drop, probably due to the introduced noise.

One additional finding is that even though the difference in Recall@2 between the original DPR and SimCSE is not large (10.1% vs 10.2%), the EM of generated answers can differ significantly (12.4% vs 9.1%). The reason may be that the retrieved content of DPR includes not only the correct answers but also other helpful information. RGFT further widens the gap of Reacll@k between DPR and SimCSE.

Table 4: The performance of RuleRAG-ICL with a powerful retriever, Contriever, under three LLMs.

| | Architecture | | RuleQA-I | | | RuleQA-Y | | | RuleQA-W | | | RuleQA-F | | | RuleQA-N | | |
|---|---|---|---|---|---|---|---|---|---|---|---|---|---|---|---|---|---|
| | Retriever | Generator | R@10 | EM | T-F1 | R@10 | EM | T-F1 | R@10 | EM | T-F1 | R@10 | EM | T-F1 | R@10 | EM | T-F1 |
| Standard RAG | Contriever | ChatGLM2_6B | 41.2 | 8.5 | 24.7 | 52.7 | 27.2 | 31.1 | 62.2 | 41.6 | 42.9 | 80.6 | 25.4 | 35.8 | 87.6 | 4.9 | 8.3 |
| RuleRAG-ICL | RG-Contriever | ChatGLM2_6B | 45.5 | 10.5 | 25.4 | 55.2 | 32.1 | 31.8 | 63.2 | 43.8 | 43.2 | 83.9 | 27.5 | 39.5 | 88.5 | 12.0 | 12.8 |
| | RG-Contriever | RG-ChatGLM2_6B | 45.5 | 10.8 | 25.6 | 55.2 | 32.9 | 32.5 | 63.2 | 46.4 | 45.9 | 83.9 | 29.6 | 40.6 | 88.5 | 16.5 | 14.2 |
| Standard RAG | Contriever | LLAMA2_7B | 41.2 | 18.7 | 36.2 | 52.7 | 41.7 | 39.6 | 62.2 | 45.5 | 51.2 | 80.6 | 42.0 | 46.1 | 87.6 | 45.2 | 56.5 |
| RuleRAG-ICL | RG-Contriever | LLAMA2_7B | 45.5 | 19.0 | 36.6 | 55.2 | 42.6 | 42.3 | 63.2 | 50.2 | 53.0 | 83.9 | 43.9 | 50.0 | 88.5 | 48.0 | 59.9 |
| | RG-Contriever | RG-LLAMA2_7B | 45.5 | 22.8 | 39.6 | 55.2 | 47.8 | 43.0 | 63.2 | 52.7 | 56.2 | 83.9 | 49.0 | 51.8 | 88.5 | 51.3 | 62.8 |
| Standard RAG | Contriever | GPT-3.5-Turbo | 41.2 | 19.1 | 27.7 | 52.7 | 38.1 | 44.2 | 62.2 | 46.5 | 43.7 | 80.6 | 56.3 | 39.1 | 87.6 | 30.7 | 59.9 |
| RuleRAG-ICL | RG-Contriever | GPT-3.5-Turbo | 45.5 | 19.7 | 30.1 | 55.2 | 41.0 | 49.9 | 63.2 | 49.4 | 65.8 | 83.9 | 56.5 | 50.3 | 88.5 | 32.6 | 64.6 |
| | RG-Contriever | RG-GPT-3.5-Turbo | 45.5 | 25.8 | 39.7 | 55.2 | 44.5 | 53.1 | 63.2 | 53.1 | 68.7 | 83.9 | 57.6 | 59.0 | 88.5 | 59.4 | 75.6 |

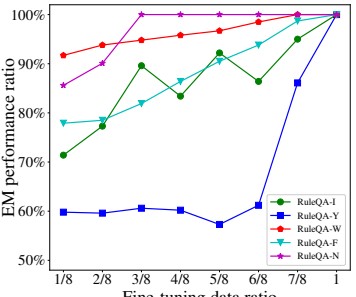

Figure 4: Due to the varying difficulty of the rules in five benchmarks, the EM performance variation of RuleRAG-FT produces different characteristics. The x-axis is the ratio of the amount of fine-tuned data to the total amount of fine-tuning data. The y-axis is the ratio of EM performance to the optimal one under DPR and LLAMA2_7B, with closer to 100% indicating stronger performance.

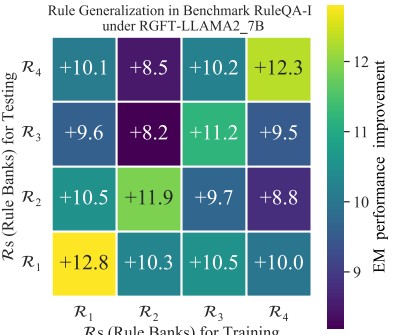

Figure 5: The EM performance of generalizing RuleRAG-FT from the source rule bank $\mathcal{R}_i$ to the target rule bank $\mathcal{R}_j$, i.e., RuleRAG-FT is trained on $\mathcal{R}_i$ and tested on $\mathcal{R}_j$. The numbers in $(\mathcal{R}_i, \mathcal{R}_j)$ represent the performance gains compared to the baseline Standard RAG tested on $\mathcal{R}_j$.

### 5.3.2 More Retrievers for RuleRAG-ICL

Contriever (Izacard et al., 2022) is a powerful retriever with strong unsupervised performance and can transfer well to new applications. Therefore, it has been widely used in RAG frameworks. In Table 4, we note that Contriever without the guidance of rules can achieve relatively good recall and RG-Contriever makes further enhancements. Compared to Standard RAG, RuleRAG-ICL with RG-Contriever and RG-generators also obtain varying degrees of performance improvement under the three LLMs. These results confirm the outstanding ability of our proposed rule-guided method.

## 6 Impact of Fine-tuning Data Volume on EM Performance

In Figure 4, the overall performance trend of the five benchmarks is that the larger the amount of fine-tuning data, the better the results. Yet, since the different properties of the rules in different benchmarks lead to different degrees of difficulty in learning, the growth of model performance under different benchmarks exhibits various characteristics. Specifically, RuleQA-F has the most intuitive growth curve: It starts from low performance and slowly grows to the optimal performance, reflecting the increasing mastery of rules during the RGFT process of RuleRAG-FT. RuleQA-W yields relatively superior performance after just one-eighth of the total amount of the fine-tuning data and further improves subsequently. RuleQA-N achieves the best EM performance after three-eighths of the fine-tuning data and maintains flat. In contrast, the performance in RuleQA-Y fluctuates modestly at a very low level throughout the first half of the RGFT process (from one-eighth to five-eighths), and then sees a sudden surge in capability during the second half of the RGFT process (from six-eighths to the end). The EM performance in RuleQA-I fluctuates more dramatically: While realizing very large EM performance gains (ranking second in all the benchmarks), it undergoes several upward and downward drops before levelling off at the optimal performance. This suggests that RuleQA-I is the most challenging among our constructed five benchmarks. Moreover, from Table 2, 3, 4, we find RuleRAG has the worst absolute performance in RuleQA-I compared to other benchmarks under the same LLMs, which also illustrates the challenge of our constructed RuleQA-I.

## 7 RULE GENERALIZATION

RuleRAG-ICL is training-free, so we can attach arbitrary rules to the method's input by in-context learning. Extensive experimental results above naturally illustrate its instruction-following ability to many kinds of rules. In the RGFT setting, the constructed fine-tuning data $\mathcal{F}_G$ for RuleRAG-FT is limited anyway but rules are inexhaustible, so RuleRAG-FT cannot and should not see the full set of rules. Therefore, it is important to verify the ability of RuleRAG-FT to generalize to untrained rules. In this experiment, RuleRAG-FT must capture the transferable rule utilization capability, since RuleRAG-FT has no prior knowledge of the target rule bank and is forced to learn from the source rule bank. The results in Figure 5, where $\mathcal{R}_i \cap \mathcal{R}_j = \emptyset$ and $|\mathcal{R}_i| = |\mathcal{R}_j|$ $(i, j \in \{1, 2, 3, 4\})$, show that (1) The diagonal $(\mathcal{R}_i, \mathcal{R}_i)$ has the highest performance gains and there are slight differences between various rule banks; (2) The results on two sides of the diagonal fluctuations within reasonable ranges and all show stable improvements over Standard RAG. This implies that RuleRAG-FT can take advantage of the ability to leverage the learned underlying rule patterns rather than being limited to concrete rule instances.

## 8 RELATED WORKS

### 8.1 RETRIEVAL-AUGMENTED GENERATION

Retrieval-augmented generation (RAG) follows the paradigm of "retrieve-then-read", where the retrieval module explicitly augments the generation module with external knowledge banks (Guu et al., 2020; Lewis et al., 2020). Retrieval approaches include sparse retrievers based on sparse bag-of-words representation (Robertson & Zaragoza, 2009), dense retrievers based on dense vectors (Karpukhin et al., 2020; Gao et al., 2021) and more complex hybrid search algorithms (Li et al., 2023a; Lin et al., 2023). The current RAG frameworks are widely adopted to complement the parametric knowledge of LLMs along different stages (Gao et al., 2024), including pre-training stage (RETRO; Borgeaud et al. (2022b), Atlas; Izacard et al. (2024), COG; Lan et al. (2023)), fine-tuning stage (Self-RAG; Asai et al. (2023), SURGE; Kang et al. (2023), CoN; Yu et al. (2023)) and inference stage (DSP; Khattab et al. (2023), KnowledGPT; Wang et al. (2023), RoG; Luo et al. (2024), CoK; Li et al. (2024)).

### 8.2 KNOWLEDGE-INTENSIVE QA

In the realm of QA, a series of queries are considered knowledge-intensive if humans or models need access to large and external corpora. Researchers have developed many systems and proven the effectiveness of RAG in many knowledge-intensive QA tasks (Petroni et al., 2021). Recently, upon the assumption that documents in the corpora can directly support the answer responses, RAFT (Zhang et al., 2024) and RA-DIT (Lin et al., 2024) fine-tune LLMs by concatenating documents and queries as prompts. However, many answers to factual queries are hidden in *semantically dissimilar but logically related events*, which leads to irrelevant information retrieved by imperfect retrievers inevitably confusing RAG methods. Therefore, the integration of rules has gained significant attention (Wang et al., 2024b;c). For instance, Wu et al. (2024) investigates the necessity of mitigating misleading irrelevant interference in RAG. Sun et al. (2024) only discusses the rule-following abilities of LLMs without retrieval and ignores how to obtain rules. In contrast, in this paper, our proposed RuleRAG involves a more comprehensive consideration of mining rules, retrieving documents and generating answers.

## 9 CONCLUSION AND FUTURE WORKS

In this paper, we point out two high-level problems of current RAG and propose a method named rule-guided retrieval-augmented generation (RuleRAG) based on observations of the objective world. RuleRAG, including RuleRAG-ICL and RuleRAG-FT, can effectively improve the performance of multiple pre-trained retrievers and generators by in-context learning and instruction fine-tuning, respectively. RuleRAG-ICL intuitively shows that RAG can directly benefit from our proposal by prompting LLMs with rules. To further improve the QA performance, RuleRAG-FT retrofits the retrievers to recall more supportive information through the designed RGFT and updates generators to make better use of the retrieved documents. Experiments show RuleRAG achieves strong performance on the five constructed rule-aware QA benchmarks. In the future, we will explore how RuleRAG can effectively retrieve and answer when facing more complex queries and adapt to a wider range of rules.

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

Table 5: The performance of RuleRAG-ICL and RuleRAG-FT for queries which may not need the guidance of rules to retrieve or generate. *The results reflect the robustness of our methods.*

| | Architecture | | RuleQA-I | | | RuleQA-Y | | | RuleQA-W | | | RuleQA-F | | | RuleQA-N | | |
|---|---|---|---|---|---|---|---|---|---|---|---|---|---|---|---|---|---|
| | Retriever | Generator | R@10 | EM | T-F1 | R@10 | EM | T-F1 | R@10 | EM | T-F1 | R@10 | EM | T-F1 | R@10 | EM | T-F1 |
| Standard RAG | DPR | LLAMA2_7B | 4.6 | 10.7 | 34.9 | 2.7 | 3.6 | 19.7 | 0.6 | 2.3 | 30.2 | 15.2 | 11.0 | 27.6 | 20.6 | 12.8 | 25.9 |
| RuleRAG-ICL | RG-DPR | RG-LLAMA2_7B | 11.8 | 11.9 | 35.5 | 5.3 | 9.5 | 23.4 | 5.9 | 2.4 | 32.5 | 26.0 | 17.0 | 39.9 | 24.9 | 17.6 | 36.3 |
| RuleRAG-FT | RGFT-DPR | RGFT-LLAMA2_7B | 39.8 | 16.6 | 36.3 | 46.8 | 28.7 | 33.8 | 34.9 | 15.9 | 34.1 | 94.1 | 35.9 | 48.9 | 33.7 | 20.4 | 37.5 |

# A APPENDIX

## A.1 THE ROBUSTNESS OF RULERAG

In the inferring process, since we can not know the content of the queries in advance, we may match some relevant rules for the queries regardless of whether the queries need the guidance of rules or not. In our preliminary experiments, we also find that, in some cases, retrieving information for some queries can directly match relevant documents.

*Therefore, in this section, we verify the robustness of our proposed method RuleRAG on queries which may not need the guidance of rules. We want to know if our introduced rules will interfere with the performance of retrieval and generation of such queries.*

Specifically, for each query in the benchmark, we degenerate it into a new relevant query by using the previously matched rules ( *[Entity 1, $r_1$, Entity 2] leads to [Entity 1, $r_2$, Entity 2]*) and ensure that the answer is unchanged and that the relevant documents can be retrieved directly from the corpus. Meanwhile, according to the principle of performance comparison, we try to minimize interference with the original queries. For instance in Figure 1, the original query is What is the nationality of Jean-Luc Godard? and the rule is that " *[Entity 1, born in, Entity 2] leads to [Entity 1, has nationality, Entity 2]*". Then, we convert the query into Where is Jean-Luc Godard born?. In this way, these queries can theoretically be successfully retrieved with related documents and correctly answered without the guidance of rules.

In order to test the robustness of our rule-guided approach RuleRAG to such queries, we first conduct the Standard RAG on them as a baseline and then test the performance of RuleRAG by adding our previously matched rules. Hence, the only difference in the input of LMs between the main experiment and this experiment is the queries. The others, including rules and answers, remain the same. The results are shown in Table 5. We find (1) In terms of absolute performance, compared Table 2, most of the results in Table 5 show a certain degree of degradation, which indicates that *we successfully achieve interference with the methods.* (2) Compared to the Standard RAG in Table 5, our proposed RuleRAG-ICL and RuleRAG-FT still achieve performance improvement over all the evaluation metrics, showing that our methods can overcome the interference of irrelevant rules. Fine-tuning based RuleRAG-FT is consistently better than RuleRAG-ICL, showing that our proposed RGFT is effective for these queries. Therefore, our methods are robust.

## A.2 THE CHOICE OF RULERAG-ICL AND RULERAG-FT

Our proposed RuleRAG includes two parts, RuleRAG-FT which requires training and RuleRAG-ICL which does not. They can also be used in combination with different LLMs: small-scale LLMs (6B, 7B, 13B in our paper) and a closed-source LLM (GPT-3.5-Turbo in our paper).

*For different usage scenarios and requirements, we are free to choose different combinations. Summarizing all the results shown in this paper, we give the following heuristic decision criteria and corresponding reasons.*

Typically, the base performance of small-scale LLMs (the baseline Standard RAG) is low and the performance improvement of both RuleRAG-ICL and RuleRAG-FT with small-scale LLMs is very significant. Therefore, we can use the RuleRAG-ICL to get good results locally when hardware resources are limited. Otherwise, we recommend fine-tuning LLMs for better results. For our benchmarks, the inference time is 3-8 hours and the time for fine-tuning with the full data is 1-3 days. If users need to get inference results quickly in a short time, we recommend calling APIs of closed-source LLMs. In this combination, our methods' absolute performance and performance

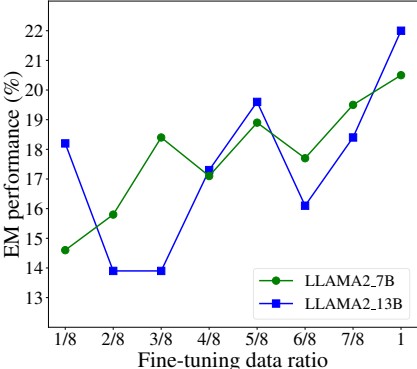

Figure 6: The EM performance of RuleRAG-FT in RuleQA-I with RGFT-LLAMA2_7B and RGFT-LLAMA2_13B under increasing fine-tuning data ratio. The retriever is kept as RGFT-DPR.

Table 6: The results of RuleRAG-ICL with Contriever under Mistral_7B_v0.2 and LLAMA2_13B.

| | Architecture | | RuleQA-I | | | RuleQA-Y | | | RuleQA-W | | | RuleQA-F | | | RuleQA-N | | |
|---|---|---|---|---|---|---|---|---|---|---|---|---|---|---|---|---|---|
| | Retriever | Generator | R@10 | EM | T-F1 | R@10 | EM | T-F1 | R@10 | EM | T-F1 | R@10 | EM | T-F1 | R@10 | EM | T-F1 |
| Standard RAG | Contriever | Mistral_7B_v0.2 | 41.2 | 12.5 | 21.3 | 52.7 | 37.8 | 36.1 | 62.2 | 43.7 | 44.9 | 80.6 | 21.5 | 36.8 | 87.6 | 30.3 | 23.3 |
| RuleRAG-ICL | RG-Contriever | RG-Mistral_7B_v0.2 | 45.5 | 15.4 | 24.5 | 55.2 | 40.8 | 39.8 | 63.2 | 46.3 | 45.8 | 83.9 | 26.1 | 39.8 | 88.5 | 39.8 | 31.9 |
| Standard RAG | Contriever | LLAMA2_13B | 41.2 | 22.1 | 39.5 | 52.7 | 40.8 | 44.2 | 62.2 | 49.2 | 54.2 | 80.6 | 42.4 | 51.4 | 87.6 | 50.2 | 57.4 |
| RuleRAG-ICL | RG-Contriever | RG-LLAMA2_13B | 45.5 | 22.3 | 39.8 | 55.2 | 41.5 | 45.8 | 63.2 | 51.2 | 52.4 | 83.9 | 46.6 | 52.2 | 88.5 | 52.7 | 58.1 |

improvement are still very high (even optimal in some cases). For our benchmarks, their inference time is 0.5-2 hours.

## A.3 THE EM PERFORMANCE TREND OF LLAMA2_7B AND LLAMA2_13B

To make a stronger argument that dataset RuleQA-I is fairly difficult, we give in Figure 6 how the EM performance of two different LLMs varies with the amount of fine-tuning dataset. From the figure, we find that the larger LLM ends up with better results (The result of LLAMA2_13B is better than LLAMA2_7B in the end), which is intuitive. LLAMA2_13B also experiences performance fluctuations, which illustrates the general challenging nature of RuleQA-I for multiple LLMs. In addition, we observe that in the second half of the fine-tuning process (the ratio from 4/8 to 1), both LLMs have similar change curves (up, then down, then up again), and the magnitude of change was greater for LLAMA2_13B than for LLAMA2_7B. We speculate that this is because both LLMs have similar model architectures, and thus the learning processes during fine-tuning are similarly guided; whereas, LLAMA2_13B has more parameters, leading to fluctuating more and ultimately performing better.

## A.4 RULERAG-ICL WITH CONTRIEVER

As a complement to the performance of RuleRAG-ICL with Contriever in Table 4, we use the other two LLMs, Mistral_7B_v0.2 and LLAMA2_13B, to show the effectiveness of our proposed rule-guided method. Table 6 shows the performance of Mistral_7B_v0.2 and LLAMA2_13B with/without rules.

## A.5 IMPLEMENTATION DETAILS

**Generator fine-tuning.** We fine-tune the ChatGLM2_6B, Mistral_7B_v0.2, LLAMA2_7B, LLAMA2_13B models using 2, 2, 4 and 8 V100 32G GPUs, respectively. We use LORA (Hu et al., 2022) with 4-bit, a parameter-efficient fine-tuning (PEFT) adaptation method, to deal with the enormous computation costs and hardware requirements in training LLM. The fine-tuning hyperparameters are detailed in Table 7. Similar to Lin et al. (2024), we find that the best generalization performance on the dev set can be achieved using a small number of fine-tuning epochs. We evaluate the models every 3 epochs and select the best checkpoint based on the average dev set performance.

Table 7: Hyperparameters for RGFT-Generators.

| LLM | lr | lora r | lora alpha | lora dropout | warm-up | batch size | epochs | model parallel | seq len |
|---|---|---|---|---|---|---|---|---|---|
| ChatGLM2_6B | 3e-5 | 4 | 16 | 0.05 | 5 | 8 | 50 | 1 | 5120 |
| Mistral_7B_v0.2 | 3e-5 | 4 | 16 | 0.05 | 5 | 8 | 50 | 1 | 5120 |
| LLAMA2_7B | 3e-4 | 8 | 32 | 0.05 | 5 | 8 | 50 | 2 | 5120 |
| LLAMA2_13B | 3e-4 | 16 | 32 | 0.05 | 10 | 4 | 50 | 4 | 5120 |

Table 8: Instruct prompt.

# Instruct: For the query in the form of "Time {time} what does {subject} {relation} ?", we provide a collection of text consisting of multiple documents in the form of "Time {time} {subject} {relation} {object}." Your response should directly generate the missing {object}.
# Retrieved documents: Documents related to the Query. Time 2014-06-23 Abdullah Abdullah Expel or withdraw peacekeepers Election Commission (Afghanistan). Time 2014-02-20 Abdullah Abdullah Make a visit Afghanistan. · · · Time 2014-07-16 Abdullah Abdullah Make a visit Ashraf Ghani Ahmadzai. · · · Time 2014-09-20 Abdullah Abdullah Make a visit Foreign Affairs (United States).
# Rules: Use the following Two rules to answer the given Query. Rule One: [Entity1, Abduct, hijack, or take hostage, Entity2] leads to [Entity1, Make a visit, Entity2]. Rule Two: [Entity1, Make a visit, Entity2] leads to [Entity1, Make a visit, Entity2].
# Query: Time 2014-12-01 what does Abdullah Abdullah Make a visit ?

# Answer: Afghanistan.

**Retriever fine-tuning.** We fine-tune DPR and SimCSE on 4 V100 32G GPUs using their public codes with a lr of 1e-5, a batch size of 32, and a temperature of 0.01. The base models are downloaded from their GitHub website.

## A.6 THE DIFFERENCE OF OUR CONSTRUCTED BENCHMARKS AND EXISTING DATASETS

Existing conventional QA datasets include HotPotQA (Yang et al., 2018), 2WikiMultiopQA (Ho et al., 2020), MuSiQue (Trivedi et al., 2022), and Bamboogle (Press et al., 2023). Although they are widely leveraged in evaluating the QA performance of LMs, we find that all these datasets are primarily focused on multi-hop and comparison-type questions and pay less attention to queries that require logical thinking to answer. As we know, many queries in the real world are not justified by relevance alone, because in many cases the lexical level of relevance is not the information that can support the answer to the query, and even introduces a lot of noise instead. Therefore, in this paper, we construct five rule-aware benchmarks based on five popular static KGs or temporal KGs to emphasize the importance of rules in the QA task. It is worth noting that our described construction way in Section 2 is general and easy to reproduce. For newly defined rule patterns, we can quickly construct corresponding benchmarks using the above construction way, showing its better scalability.

Moreover, our constructed benchmarks also provide corresponding fine-tuning datasets, which aim to improve the retrieval and generation ability of LMs. Currently, obtaining high-quality and plentiful supervised data for a specific task is a challenging problem for researchers (Wang et al., 2024a). Manual annotation is time-consuming and difficult to replicate. A very convenient and widely used way is to distil knowledge from LLMs. However, relying on LLMs to generate data for *training* puts too much trust in them and does not actually guarantee the accuracy of the reasoning ability in the trained models.

*In contrast, in this paper, the fine-tuning datasets of the retrievers are obtained by pattern matching and retrieval recall; the fine-tuning datasets for generators are obtained by using the KG nodes as answers and using retrieved information as instructions. The entire process is efficiently streamlined and automatically generated.*

## A.7 CASE STUDY

A concrete example in Table 9 visually compares the baseline model (Standard RAG) and our proposed methods, RuleRAG-ICL and RuleRAG-FT.

Table 9: A detailed case study in RuleQA-I. We show the retrieved documents of three kinds of retrievers (DPR, RG-DPR, RGFT-DPR) and the answers of Standard RAG, RuleRAG-ICL and RuleRAG-FT with LLAMA2_13B.

| |
|---|
| **Query**:
Time 2014-12-11 what does Court Judge (Nigeria) Accuse ? |
| **Ground Truth**:
Citizen (Nigeria). |
| **Rules**:
Rule One:
[Entity1, Accede to demands for change in leadership, Entity2] leads to [Entity1, Accuse, Entity2].
Rule Two:
[Entity1, Ease administrative sanctions, Entity2] leads to [Entity1, Accuse, Entity2].
Rule Three:
[Entity1, Appeal for diplomatic cooperation, Entity2] leads to [Entity1, Accuse, Entity2]. |
| **Retrieved documents of DPR (top-10)**:
① Time 2014-08-22 Representatives (Nigeria) Consult Media (Africa). ② Time 2014-05-27 Activist (Nigeria) Consult Associated Press. ③ Time 2014-06-16 Education (Nigeria) Consult Gabriel Torwua Suswam. ④ Time 2014-09-03 Media (Nigeria) Consult Stephen Davis. ⑤ Time 2014-05-21 Media (Nigeria) Consult Ministry (Nigeria). ⑥ Time 2014-09-03 Media (Nigeria) Consult Stephen Davis. ⑦ Time 2014-08-29 Media (Nigeria) Consult Stephen Davis. ⑧ Time 2014-03-19 Citizen (Nigeria) Accuse Media (Nigeria). ⑨ Time 2014-05-27 Activist (Nigeria) Consult Associated Press. ⑩ Time 2014-10-07 Amnesty International Criticize or denounce Representatives (Nigeria). |
| **Retrieved documents of RG-DPR (top-10)**:
① Time 2014-11-06 Court Judge (Nigeria) Arrest, detain, or charge with legal action Boko Haram. ② Time 2014-07-07 Court Judge (Nigeria) Make optimistic comment Nigerian Bar Association. ③ Time 2014-10-01 Court Judge (Nigeria) Arrest, detain, or charge with legal action Boko Haram. ④ Time 2014-06-12 Court Judge (Nigeria) Arrest, detain, or charge with legal action Citizen (Nigeria). ⑤ Time 2014-07-21 Court Judge (Nigeria) Arrest, detain, or charge with legal action Citizen (Nigeria). ⑥ Time 2014-04-11 Court Judge (Nigeria) Arrest, detain, or charge with legal action Citizen (Nigeria). ⑦ Time 2014-08-26 Court Judge (Nigeria) Appeal for diplomatic cooperation (such as policy support) Citizen (Nigeria). ⑧ Time 2014-04-04 Court Judge (Nigeria) Appeal for diplomatic cooperation (such as policy support) Government (Nigeria). ⑨ Time 2014-09-16 Court Judge (Nigeria) Appeal for diplomatic cooperation (such as policy support) Citizen (Nigeria). ⑩ Time 2014-07-08 Court Judge (Nigeria) Make optimistic comment Nigerian Bar Association. |
| **Retrieved documents of RGFT-DPR (top-10)**:
① Time 2014-09-16 Court Judge (Nigeria) Appeal for diplomatic cooperation (such as policy support) Citizen (Nigeria). ② Time 2014-04-03 Court Judge (Nigeria) Appeal for diplomatic cooperation (such as policy support) Other Authorities / Officials (Nigeria). ③ Time 2014-08-26 Court Judge (Nigeria) Appeal for diplomatic cooperation (such as policy support) Citizen (Nigeria). ④ Time 2014-04-04 Court Judge (Nigeria) Appeal for diplomatic cooperation (such as policy support) Citizen (Nigeria). ⑤ Time 2014-01-22 Court Judge (Nigeria) Ease administrative sanctions Citizen (Nigeria). ⑥ Time 2014-09-16 Court Judge (Nigeria) Express intent to cooperate Citizen (Nigeria). ⑦ Time 2014-07-17 Court Judge (Nigeria) Ease administrative sanctions Citizen (Nigeria). ⑧ Time 2014-02-17 Court Judge (Nigeria) Ease administrative sanctions Member of Legislative (Govt) (Nigeria). ⑨ Time 2014-02-28 Court Judge (Nigeria) Make an appeal or request Citizen (Nigeria). ⑩ Time 2014-08-11 Court Judge (Nigeria) Make an appeal or request Citizen (Nigeria). |
| **Answer of Standard RAG (DPR + LLAMA2_13B)**:
Media (Africa). |
| **Answer of RuleRAG-ICL (RG-DPR + RG-LLAMA2_13B)**:
Citizen (Nigeria). |
| **Answer of RuleRAG-FT (RGFT-DPR + RGFT-LLAMA2_13B)**:
Citizen (Nigeria). |

Specifically, the documents retrieved by the original DPR are almost irrelevant to the query and only one out of the top 10 documents contains the correct answer "Citizen (Nigeria)". RG-DPR's retrieval results are more relevant to the query entity and semantically support the answer. Meanwhile, 5 of the top 10 documents contain the correct answer. The retrieval quality of the fine-tuned RGFT-DPR is the best. All the retrieved documents are strongly supportive while answering the query through the given rules. In addition, 8 out of the top 10 documents contain correct answers, which further reflects the strong performance of our proposed methods.

Moreover, in the answering stage, Standard RAG naturally obtains a wrong answer based on low-quality retrieval results. However, RuleRAG-ICL and RuleRAG-FT attribute the correct answer through in-context learning and fine-tuning under the guidance of the rules.

### A.8 ERROR ANALYSIS

We further analyzed the detailed performance of our proposed model on 60*5 incorrectly answered queries from the five benchmarks. There were three main classes of errors:

(a) Rule Failure (5%): In the real world, rules can reflect the logical workings of most events. However, we cannot claim that absolutely no exceptions occur. Among the incorrect responses we sampled, we found that the answers to some questions did not follow the general rules of reasoning, which in turn resulted in response failures. Future work could address such special cases separately.

(b) Retrieval Error (55%): In this section, we assume that a retrieval is considered correct as long as the correct answer is included in the top 10 recalled documents, and a retrieval is considered incorrect otherwise. Due to the very large size of the corpus and the large number of documents that are semantically similar but do not support the answer, even a fine-tuned retriever may not recall relevant facts for the correct answer. In almost all cases, the question can not be answered correctly if the retrieved documents are wrong.

(c) Attribution Error (40%): Due to the complex logical relationships between events, when the retrieved documents contain the correct answer, the generator may still fail to follow the rules and then come up with an incorrect answer. Generally, the more documents in the top 10 retrieved information that are related to the correct answer, the higher the probability that the generator will answer correctly. The problem of attribution error occurs generally because there are only one to three supportive documents in the retrieved information.

### A.9 PROMPT TEMPLATES

There are mainly two kinds of prompts in our model: prompts for fine-tuning in Figure 2 and prompts for in-context learning of GPT in Table 4. As Figure 2 shows, Instruct prompts consist of five parts: *Instruct*, *Retrieved documents*, *Rules*, *Query* and *Answer*. The *Instruct* is fixed, the *Retrieved documents* are retrieved by our proposed RuleRAG according to *Rules* and *Query*, and the *Answer* is pre-defined. As Section 4 shows, we use 3-shot in-context learning for GPT to replace fine-tuning. In the following, we take RuleQA-I as an instance to show the RGFT instruct prompts (Table 8) and prompts for GPT-3.5-Turbo (Table 10).

Table 10: GPT-3.5-Turbo prompt.

Answer the Final Query by referring to the three cases below.

Case 1:
# Instruct: For the query in the form of "Time {time} what does {subject} {relation} ?", we provide a collection of text consisting of multiple documents in the form of "Time {time} {subject} {relation} {object}." Your response should directly generate the missing {object}.
# Retrieved documents: Documents related to the Query. Time 2014-06-23 Abdullah Abdullah Expel or withdraw peacekeepers Election Commission (Afghanistan). Time 2014-02-20 Abdullah Abdullah Make a visit Afghanistan. ··· Time 2014-07-16 Abdullah Abdullah Make a visit Ashraf Ghani Ahmadzai. ··· Time 2014-09-20 Abdullah Abdullah Make a visit Foreign Affairs (United States).
# Rules: Use the following Two rules to answer the given Query. Rule One: [Entity1, Abduct, hijack, or take hostage, Entity2] leads to [Entity1, Make a visit, Entity2]. Rule Two: [Entity1, Make a visit, Entity2] leads to [Entity1, Make a visit, Entity2].
# Query: Time 2014-12-01 what does Abdullah Abdullah Make a visit ?
# Answer: Afghanistan.

Case 2:
# Instruct: For the query in the form of "Time {time} what does {subject} {relation} ?", we provide a collection of text consisting of multiple documents in the form of "Time {time} {subject} {relation} {object}." Your response should directly generate the missing {object}.
# Retrieved documents: Documents related to the Query. Time 2014-04-07 Adams Oshiomhole Make an appeal or request Citizen (Benin). Time 2014-10-13 Adams Oshiomhole Accuse People's Democratic Party (Benin). ··· Time 2014-07-02 Adams Oshiomhole Criticize or denounce Citizen (Nigeria). ··· Time 2014-08-05 Adams Oshiomhole Praise or endorse Labor Union (Nigeria).
# Rules: Use the following Three rules to answer the given Question. Rule One: [Entity1, Make an appeal or request, Entity2] leads to [Entity1, Make an appeal or request, Entity2]. Rule Two: [Entity1, Appeal for economic aid, Entity2] leads to [Entity1, Make an appeal or request, Entity2]. Rule Three: [Entity1, Accuse of aggression , Entity2] leads to [Entity1, Make an appeal or request, Entity2].
# Query: Time 2014-12-01 what does Adams Oshiomhole Make an appeal or request ?
# Answer: Citizen (Nigeria).

Case 3:
# Instruct: For the query in the form of "Time {time} what does {subject} {relation} ?", we provide a collection of text consisting of multiple documents in the form of "Time {time} {subject} {relation} {object}." Your response should directly generate the missing {object}.
# Retrieved documents: Documents related to the Query. Time 2014-09-25 Adams Oshiomhole Demand Citizen (Benin). Time 2014-02-05 Adams Oshiomhole Express intent to cooperate Citizen (Nigeria). ··· Time 2014-10-13 Adams Oshiomhole Make an appeal or request Other Authorities / Officials (Nigeria). ··· Time 2014-07-01 Adams Oshiomhole Praise or endorse Media (Africa).
# Rules: Use the following Three rules to answer the given Question. Rule One: [Entity1, Obstruct passage, block, Entity2] leads to [Entity1, Praise or endorse, Entity2]. Rule Two: [Entity1, Expel or deport individuals, Entity2] leads to [Entity1, Praise or endorse, Entity2]. Rule Three: [Entity1, Praise or endorse , Entity2] leads to [Entity1, Praise or endorse, Entity2].
# Query: Time 2014-12-01 what does Adams Oshiomhole Praise or endorse ?
# Answer: Media (Africa).

Final Query:
# Instruct: For the query in the form of "Time {time} what does {subject} {relation} ?", we provide a collection of text consisting of multiple documents in the form of "Time {time} {subject} {relation} {object}." Your response should directly generate the missing {object}.
# Retrieved documents: Documents related to the Query. Time 2014-03-11 Alexis Tsipras Make a visit Ireland. Time 2014-02-26 Alexis Tsipras Express intent to meet or negotiate Slovenia. ··· Time 2014-05-26 Alexis Tsipras Make a visit Head of Government (Greece). ··· Time 2014-09-17 Alexis Tsipras Consult New Democracy.
# Rules: Use the following Three rules to answer the given Question. Rule One: [Entity1, Accede to demands for change in leadership, Entity2] leads to [Entity1, Make statement, Entity2]. Rule Two: [Entity1, Demand release of persons or property, Entity2] leads to [Entity1, Make statement, Entity2]. Rule Three: [Entity1, Accuse of crime, corruption , Entity2] leads to [Entity1, Make statement, Entity2].
# Query: Time 2014-12-01 what does Alexis Tsipras Make statement ?
# Answer:

