# OpenReview forum: "RuleRAG: Rule-Guided Retrieval-Augmented Generation with Language Models for Question Answering"
_ICLR.cc/2025/Conference — Submitted to ICLR 2025_

### Official Review · Reviewer_V8nK · 2024-10-31

**Soundness:** 3
**Presentation:** 4
**Contribution:** 3
**Rating:** 6
**Confidence:** 4

**Summary:**

The paper presents RuleRAG, an innovative approach that combines rule-guided retrieval and generation for knowledge-intensive question answering. This method addresses the limitations within existing RAG frameworks by utilizing symbolic rules to direct both retrievers and generators. RuleRAG enhances performance through in-context learning and fine-tuning processes. Additionally, it establishes rule-aware QA benchmarks and shows substantial improvements over standard RAG across various metrics.

**Strengths:**

1. RuleRAG uniquely combines rule-based guidance with retrieval and generation, showing promising results in knowledge-intensive QA.
2. RuleRAG has the capability to generalize beyond the rules it has been specifically trained on, albeit with less than optimal performance. This ability highlights its significant advantage and potential for broader applications beyond its original rule set.
3. The construction of five rule-aware QA benchmarks, including both temporal and static scenarios, provides a thorough evaluation of RuleRAG's capabilities and its scalability across different QA tasks.
4. RuleRAG shows significant improvement over standard RAG, and the experimental results appear solid.

**Weaknesses:**

1. The performance of the RuleRAG framework is heavily dependent on the quality and coverage of the mined rules, which may not always be comprehensive or accurate. The effectiveness of RuleRAG hinges on the accuracy of the rule-mining process; if the algorithms used (AMIE3 for static KGs and TLogic for temporal KGs) fail to extract high-confidence rules, the performance of the entire framework could be compromised.
2. The paper highlights RuleRAG's potential vulnerability to irrelevant or misleading rules, a critical issue for question-answering systems using external knowledge. It lacks detailed mechanisms for filtering such rules, risking suboptimal performance during retrieval and generation phases. The authors could consider implementing or evaluating specific filtering mechanisms, such as incorporating a rule relevance scoring step, or exploring the feasibility of using the LLM itself to assess rule applicability prior to retrieval.
3. This paper emphasizes individual rules and their impact on QA performance but does not explore the interactions between multiple rules or complex rule hierarchies, which are essential for handling more sophisticated queries.

**Questions:**

1. How does RuleRAG handle exceptions or anomalies, particularly for domain-specific cases (e.g., outliers in temporal data or unrecognized entities) that do not conform to the predefined rules?
2. Given that the quality and coverage of these rules can significantly impact overall performance, what specific strategies can be implemented to ensure the diversity and representativeness of the rules extracted from the knowledge graph? For example, how can the authors incorporate techniques to evaluate rule completeness or prioritize rules based on domain relevance?
3. At times, the retrieved content might not directly include answers or rules, but still plays a crucial role in helping an LLM understand and respond to inquiries. How does RuleRAG ensure that this valuable contextual information is retained, particularly in cases where the relevant content may be ambiguous or indirect?

---

### Official Review · Reviewer_JFxQ · 2024-11-02

**Soundness:** 3
**Presentation:** 3
**Contribution:** 2
**Rating:** 3
**Confidence:** 3

**Summary:**

This paper introduces a novel framework for improving knowledge-intensive question answering (QA) by integrating symbolic rules into the Retrieval-Augmented Generation (RAG) paradigm. The authors identify two main limitations of standard RAG models: 1) insufficient retrieval relevance, as retrievers often fail to capture logical relationships, and 2) lack of explicit guidance for language models on how to utilize retrieved documents.

To address these, RuleRAG incorporates symbolic rules to guide both the retrieval and generation phases, enhancing answer accuracy. The framework operates in two modes: **RuleRAG-ICL** (In-Context Learning), which uses rules to steer retrieval and generation in a training-free manner, and **RuleRAG-FT** (Fine-Tuning), which fine-tunes models to strengthen rule adherence. Additionally, the authors develop five rule-aware QA benchmarks to test the system across temporal and static knowledge scenarios. Experimental results show effective improvements in retrieval relevance (Recall@10) and answer accuracy (Exact Match) compared to standard RAG, showcasing RuleRAG's ability to generalize to unseen rules and scale effectively.

**Strengths:**

1. The paper conducted extensive experiments and analyses to validate the effectiveness of the proposed method, despite some biases in the experimental setup.
2. The motivation for introducing rules to enhance the utilization of documents is reasonable, and experiments have shown that this method indeed helps improve the model's performance on the evaluation datasets compared to direct RAG.

**Weaknesses:**

1. The evaluation set is not robust enough: The method's self-constructed dataset seems to favor scenarios where rule-based approaches are required to answer correctly, which introduces some bias. The authors did not evaluate the method on more widely and commonly used benchmarks such as NQ, TQ, HotpotQA, StrategyQA, etc.

2. The baseline models are not comprehensive: Comparing only with direct RAG seems somewhat weak. Currently, there are more variants of RAG, such as decomposing questions before retrieval, which can also address multi-hop retrieval to some extent.

3. Limited generalizability: The paper does not conduct experiments on a broader range of datasets, making it difficult to demonstrate the method's generalizability, especially in scenarios where large models have been fine-tuned.

**Questions:**

See the weakness.

---

> ### Author Response · Authors · 2024-11-22
> **Looking forward to a discussion before the deadline**
>
> Dear Reviewer,
>
> Thanks again for your reviewing our paper!
>
> It seems that your main concern is the lack of experimental results on general QA datasets,which leads to your conclusion stating our framework is not general. However, we want to emphasize again that this paper is not intended to outperform previous approaches on common QA tasks. Instead, our contribution lies in proposing a new RAG-based framework and proving its effectiveness in scenarios where rules are available. We do not expect our approach to achieve superior performance on datasets like NQ and HotpotQA, where such rules are not applicable.
>
> As the deadline for the discussion is fast approaching, we are really looking forward to having a discussion with you on the OpenReview system. Would you mind checking our response and letting us know if you have further questions?
>
> With sincere regards,
> Authors of Paper 689

---

> ### Comment · Reviewer_JFxQ · 2024-11-29
> **Response to the Authors**
>
> Thank you for the author's response. However, the reply did not address my concerns regarding limited innovation. Additionally, it seems that the author may have violated the anonymity policy when responding to other reviewers. Therefore, I will maintain my original score.

---

> > ### Author Response · Authors · 2024-11-29
> > **Official Comment from Authors**
> >
> > Dear Reviewer JFxQ
> >
> > We would like to extend our thanks to you for your careful and constructive feedback on our manuscript. Your time and effort in reviewing our work are deeply appreciated.
> >
> > We are in the process of revising the manuscript and the accompanying code, incorporating your feedback to ensure the work is more robust and comprehensive.
> >
> > We truly value your contribution to this process and are grateful for your thoughtful input.
> >
> > Sincerely,
> >
> > Paper 689 Authors

---

### Official Review · Reviewer_n6BG · 2024-11-02

**Soundness:** 3
**Presentation:** 3
**Contribution:** 3
**Rating:** 5
**Confidence:** 3

**Summary:**

This paper proposed RuleRAG, a new approach to improve the performance of Retrieval-Augmented Generation (RAG) systems. Specifically, RuleRAG takes symbolic rules from the knowledge graph and introduces them into the retriever and generator. The proposed approach consists of two versions: RuleRAG-ICL (context learning) and RuleRAG-FT (fine-tuning), which have shown significant performance gains in several benchmarks.

**Strengths:**

1. This paper proposes a graph augmented generation framework, RuleRAG, aimed at improving the RAG system by incorporating entity relationships from the knowledge base.
2. The authors clearly outline their approach, including both RuleRAG-ICL and RuleRAG-FT. Details of the retrievers and generators are comprehensively explained, along with an in-depth description of the dataset construction.
3. New rule-aware benchmarks were created to evaluate the proposed method, and extensive experiments demonstrate the method’s effectiveness on these benchmarks .

**Weaknesses:**

1. This paper proposes RuleRAG and claims that it uses a rule-guided method to build a RAG framework for question and answer tasks. However, the experiments conducted are not comprehensive enough: this paper tests the performance of the proposed method only on a self-constructed RuleQA-series dataset, which seems to be a KBQA task in disguise . Many representative RAG tasks such as NQ, TriviaQA (for short-form QA), ASQA ( for long-form QA), HotPotQA (for multi-hop QA) are not tested, which leads us to not be able to fairly observe the performance of RuleRAG in different scenarios.


2. The RuleQA dataset proposed in this paper is constructed based on entities in the KB. However, the rules in the rule base, the queries in the test dataset and the documents in the corpus are also derived from the KB, does it mean that for every query in RuleQA, there exist exactly corresponding rules and documents to answer the question? Such a construction seems to be more favorable for RuleRAG, as it makes RuleRAG have correspondences for acquiring and introducing rules, however this is not possible in real-world QA. The authors should conduct experiments on a wide range of publicly credible datasets to validate the effectiveness of their approach.

3. Some methods, such as IRCOT, Self-RAG and GraphRAG, are not  compared. They also focus on extracting more accurate query-related contents for building a more effective RAG system.

4. The backbone retriever is not new. Some methods, such as ANCE and BGE, are not compared.

**Questions:**

1. How to control and estimate the quality of the constructed dataset?
2. The dataset link should also be anonymous.

---

### Official Review · Reviewer_RTdS · 2024-11-04

**Soundness:** 3
**Presentation:** 3
**Contribution:** 3
**Rating:** 6
**Confidence:** 3

**Summary:**

This paper presents a novel method that incorporates the rules from the existing knowledge base to aid the retrieval augmented generation. The rules are included in both retrieval and generation stage where the improvements are quite significant compared to the vanilla approach. The authors conduct extensive experiments with state-of-the-art LLMs and demonstrate the generalization of their approach.

**Strengths:**

1. This paper presents a novel research direction to add rules from knowledge base to help question answering.
2. The experiments are solid with many existing SOTA models with convincing results.
3. The authors also demonstrates the generalization of their rules.
4. The code is open sourced which will help the community.

**Weaknesses:**

1. Which mining method is the best or how to choose the mining method is missing in the paper

**Questions:**

Why do the authors not choosing some of the SOTA decoder-only retrievers, such as E5-mistral-7b-instruct? Will it better help the ICT compared to the BERT based methods, e.g. DPR, contriever?

---

### Official Review · Reviewer_vHHJ · 2024-11-04

**Soundness:** 2
**Presentation:** 3
**Contribution:** 2
**Rating:** 5
**Confidence:** 3

**Summary:**

This paper designed a retrieval-augmented generation framework named RuleRAG to address the limitations of existing RAG approaches in knowledge-intensive question answering. RuleRAG leverages symbolic rules to guide the retrieval and generation processes
 to ensure that retrieved documents are logically relevant to the query and the generated answers properly refer to the retrieved documents. To validate the effectiveness of the proposed method, the paper introduces five new QA benchmarks that require reasoning and utilize rules based on existing benchmarks. Experimental results show that RuleRAG achieves strong performance.

**Strengths:**

- **Neural symbolic method**: RuleRAG explicitly incorporates symbolic rules into a neural language model, providing clear guidance for the retrieval stage and ensuring that the generated answers are logically consistent with the retrieved information.
- **Comprehensive design and evaluation**: RuleRAG-ICL and RuleRAG-FT are designed for different scenarios, and both demonstrate strong generalization capabilities with various LLMs and retrievers.

**Weaknesses:**

- **Possibly limited scope of application**: I'm not sure whether the symbolic rules could be applied to real-world applications, since the motivating example is somewhat straightforward.
- **Limited rule learning method**: Obtaining high-quality rules is non-trivial. This paper leverages some rule induction tools like AMIE on structured knowledge resources, and what if unstructured text?
- **Unclear evaluation benchmark**: I do not understand why to construct evaluation data based on these benchmarks, which are not designed for the knowledge-intensive task. Why not consider more popular complex KBQA benchmarks, e.g., HotpotQA? On the other hand, these benchmarks may be seen when model training. Such choices may influence the convincingness of the conclusions.

**Questions:**

- **About the retrieval and generation**: I'm not sure whether the mappings between rules and retrieved results are recorded and used in the generation.
- **About the conclusions in Section 5.2**: The second conclusion (i.e., the introduced rules can provide better guidance when using larger models with the same LLM architecture) and the third one (RGFT is fairly effective and necessary for lightweight LLMs) are somewhat contradictory.

---

### Meta-Review · Area_Chair_s2kX · 2024-12-08

**Metareview:**

The paper proposes Rule-Guided Retrieval-Augmented Generation (RuleRAG) with LMs, which explicitly introduces symbolic rules to guide retrievers to retrieve logically related documents. RuleRAG can be instantiated as both in-context learning or supervised fine-tuning. While the overall framework is interesting and promising, the reviewers believe that the evaluations as of now are insufficient to validate the generalization and robustness of the proposed method. The authors should consider these points when revising their manuscript.

**Additional Comments On Reviewer Discussion:**

The reviewers reached a consensus on rejection.

---

### Decision · Program_Chairs · 2025-01-22

Reject